# Contribution of Accessibility to Urban Resilience and Evacuation Planning Using Spatial Analysis

**DOI:** 10.3390/ijerph20042913

**Published:** 2023-02-07

**Authors:** Meng-Ting Tsai, Hung-Wen Chang

**Affiliations:** Department of Architecture, National Taiwan University of Science and Technology (NTUST), Taipei City 106335, Taiwan

**Keywords:** space syntax, visible, axial, geometric accessibility, spatial analysis, efficiency evacuation route

## Abstract

Cities evolve and change with economic development and population growth, and urban planning laws in Taiwan have regulations that should be comprehensively reviewed every six years. Most current government policies aim to add new disaster prevention shelters or rescue stations. An economical way to improve the disaster prevention capabilities of urban planning is through examining or reviewing spatial structures and disaster prevention plans from the perspective of citizens or residents. The UN Office of Disaster Risk Reduction (UNDRR) launched the Making Cities Resilient Campaign policy to enhance disaster-resistant and sustainable urban environments through integrated disaster mitigation, reduction, response, and evacuation plans. This study used space syntax to ascertain evacuation route features using geometric distance analysis. There was 31.61% efficiency in relation to accessible roads on a comprehensive map. We could clearly see that since the areas in the first quadrant were relatively close to accessible roads, and there was an area that was not connected to the existing evacuation channels. The increased number of channels was more accessible and extensive. Such suggestions are helpful for government departments to prepare for disaster management. The spatial characteristics of the physical environment are explained by the accessibility and efficiency of axial maps and visibility analyzed by space syntax. Our findings reveal that space syntax is an important application when examining evacuation maps.

## 1. Introduction

The benchmark of a vibrant city is not only whether its signature buildings are successful in their purpose, the functions of land-use zoning are organized, or the civil facilities are stable but also whether urban security is satisfied. Issues pertaining to sustainability and resilience have become an important trend in spatial analysis literature [1,2]. All cities should be built on the basis of urban safety [3]. In 2010, the UN Office of Disaster Risk Reduction (UNDRR) launched a policy, the Making Cities Resilient Campaign [4], to enhance disaster-resistant and sustainable urban environments via integrated disaster mitigation, reduction, response, and evacuation plans. Evacuation plans are especially critical in fulfilling the eighth guideline—“Increase infrastructure resilience”—of the UNDRR guidebook. In Japan, Taiwan, China, and South Korea, urban planning rules are all aimed at minimizing the loss of life and property in future disasters.

In Taiwan, earthquakes are highly destructive natural disasters causing serious damage, such as the collapse of houses, civil infrastructure damage, and obstruction of rescue actions. Therefore, rapid accessibility and effective evacuation maps are important goals for governments, planners, and even ambulance corps to pursue. Currently, disaster prevention and evacuation maps are planned for all regions of Taiwan. Evacuation maps also inform residents of the location of shelters and evacuation routes. Currently, most of the academic research on evacuation routes is divided into two fields. The first is from the perspective of planning to manage disaster risk and preparing for disaster response. The second is evaluating reliability and structural resistance to ensure the safety of houses, roads, and bridges. However, the former lacks rigorous quantitative theoretical support, and the latter lacks a holistic perspective [5]. A solution is to reinforce the street networks as the backbone of urban space development and construction supporting the flow of people and vehicles through basic channels. This plays a critical and core role in urban disaster evacuation and rescue plans [6]. Poor accessibility of urban road networks leads to decreased efficiency of evacuation as well as the rescue teams accessing those in need. This poses a serious challenge to the sustainable development of an environment incorporating urban resilience. On the other hand, accessibility reflects the relative value of a certain location within a city and the convenience of social and transport activities from the perspective of urban structure and social development [7].

Cities evolve and change with economic development and population growth, and urban planning laws in Taiwan have regulations that should be comprehensively reviewed every six years. Most of the current government policies are focused on adding new disaster prevention shelters or rescue stations. An economical way to improve the disaster prevention capabilities of urban planning is through examining or reviewing spatial structure and disaster prevention plans from the perspective of citizens or residents. The evacuation plans devised for Taipei city provide people with basic information on voluntary evacuation and maps indicating safe paths to shelters [8]. This is the same as in Japan. Therefore, this work studied, a village evacuation map (Figure 1) for a local area and a district prevention map (Figure 2) for a broader area based on the width of roads as a key point for urban planners. To realize spatial efficiency beyond the point of view of a planner, a digital analysis tool, space syntax, was used in this study with topology indicators to evaluate the accessibility and efficiency of axial maps and visibility and road networks’ capacity. The results of a case study of the Nei-Hu District were beneficial in developing this issue and keeping people safe when evacuating.

This paper discusses the rational thinking of disaster prevention and evacuation mapping, using space syntax theory, to establish research evaluation factors and quantitative analysis tools for the accessibility of disaster evacuation roads and urban structures. Further, this study aimed to establish the relationship between urban configuration and big data for disaster management. Therefore, the accessibility level of a road network is an important factor affecting the sustainable development of resilient cities. The focus of this study was to enable people to immediately escape or quickly find a rescue path in the event of a disaster if buildings collapse or in the event of vulnerability or damage. In the future, the greatest degree of accessibility will provide new methods and ideas for urban disaster prevention and evacuation road network planning.

## 2. Materials and Methods

### 2.1. Resilience Theory and Space Syntax

Resilience theory is increasingly used as an approach for understanding the non-linear dynamics of “social–ecological” systems. In social–ecological systems, natural environment systems and human systems are integrated into the binary opposition of human–environment [9].

This study transforms knowledge of resilience, disaster prevention planning, and urban safety systems services into spatial patterns that can be applied to practice in urban design. Therefore, space syntax was used to solve the micro-scale urban design problems in this study.

Space syntax is a method of analyzing street networks as spaces of pedestrian movement, and its objective is to obtain knowledge of supporting architecture and urban design [2,10]. An understanding of “urban space perception” (i.e., the streets where people experience the city) has not been developed completely because it merely involves an analysis of the macro level for city systems [11].

Based on space syntax studies, strong correlations between global properties of the urban grid and pedestrian movement patterns are commonly found [12,13]. In street networks, streets that can be easily reached and are clustered with plenty of activities are important places for retail frontage [14,15]. Thus, the perceived sensory dimensions of urban green spaces are emphasized [16].

Space syntax analysis of the city has made major contributions to understanding cities’ spatial structure. To understand the bottom-up evolution of complexity in city systems (i.e., a nonlinear correlation system between variables that at higher spatial scales would be affected by the change of lower ones unexpectedly), the model of urban space perception was developed and analyzed [11]. To intervene in city systems based on experience using urban design (the spatial patterns are the main final medium), knowledge of the details of variables and spatial patterns is required. Thus, the analytical theory has been applied to the practice of urban planning successfully.

The main idea of geographic descriptions and measurements is to capture the variables that are not involved in the analysis of space syntax, such as density and diversity [17]. In addition, the concept of modeling the city building according to the urban space perception has been developed and “intensely extended”. The model is based on the concept of a spatial network of cities. Thus, the spatial capital in terms of use value and exchange value is formed by the correlations among different city locations.

### 2.2. Geometric Representation and Literature Review

Accessibility analysis of urban disaster planning reflects the spatial imbalance of rescue system supply and residents’ evacuation [18,19]. There are many methods to measure and rank the efficiency of accessibility. Makrí and Folkesson compared existing accessibility measures for a comprehensive analysis of land use impact on accessibility using a GIS [20]. Gravity-based measurement is a common method. Therefore, Geertman and van Eck [21] modified the gravity-based model and developed a measure with a meaningful unit to evaluate the aggregate accessibility at a site.

Recently, there has been a growing interest in accessibility analysis based on space syntax [22,23,24]. Different from gravity-based measures, the space syntax accessibility measure uses topological distance, not metric distance [25]. Space syntax provides a view of geometric accessibility, not geographic accessibility. The tool of space syntax analysis relies on the invention of the axis diagram, which represents urban space constituted by urban forms from the perspective of cognitive subjects (that is, people who experience and act) [26]. An axis map consists of a minimum number of lines covering all accessible urban spaces within the area of analysis, where each line on the map (referred to as an axis) represents a visually and physically accessible urban space. Thus, an axonometric diagram captures the phenomenological aspects of urban space through a single axis and the systemic aspects of urban space through the configuration of very simple geometric shapes in a complete axonometric diagram [27]. This forms a basic geometric representation on which different calculations can be performed, such as the topological accessibility with each axis in the map, which measures the so-called integral value of each line. In a series of studies worldwide, this analysis showed a strong correlation between this combined value and pedestrian movement, the most generic aspect of urban life [2].

### 2.3. Urban Disaster Evacuation Planning and Literature Review

Taiwan city governance urban planning laws, including disaster prevention plans, must be comprehensively reviewed every six years. Most of the current government policies aim to add new disaster prevention shelters or rescue stations instead of reviewing the spatial structure, and disaster prevention plans are an economical way to improve disaster prevention ability. The efficiency of urban road networks may change during city growth or urban renewal. Some studies have used geographical distance to enhance a disaster response’s evacuation capacity. Tsukaguchi and Li applied a discriminative model to verify the causes of road closures after the Great Hanshin-Awaji Earthquake [28]. They developed a simulation model to improve different street structures for the city route network. Odani and Uranaka analyzed traffic conditions immediately after the Great Hanshin-Awaji Earthquake, considering the main roads and other minor roads that suffered serious damage [29]. Lee and Yeh surveyed after the 921 Great Earthquake and found that a street width of less than 4 m was the main reason for road closure after earthquakes in Taiwan [30]. Chen et al. combined reliability and uncertainty analysis, network equilibrium models, and sensibility analysis of an equilibrium network flow to assess the performance of a degradable road network [31]. As described, Leu and Hou also performed a successful evacuation route analysis based on geographical distance. Leu Chiang-Hui proposed a route selection model in an urban earthquake disaster relief study to improve transportation efficiency by 33% [32]. Hou Peng-Hsi established a survivable network design model for earthquake disasters to decrease the average travel cost by 33% [33].

The adoption of the geometric distance study approach to conducting various hazard modeling studies appears to offer innovative opportunities for enhanced conversation in numerous fields. Furthermore, several studies have shown that using geometric layout analysis can connect other complementary information and effectively help the government better prepare, plan for, and respond to various unpredictable natural disasters at the city level. Srinurak used configuration analysis to develop integrated urban disaster prevention and mitigation strategies [34], while Mohareb used its application to improve ongoing evacuation simulation models [35]. In addition, Chang and Lee utilized configuration analysis to assess the geometric distance of emergency shelters for effective disaster response [36], and Penchev utilized previous studies to develop post-disaster stratified intervention strategies to minimize indirect impacts and promote continuity of economic activity in disaster-stricken areas [37]. These authors addressed the problem of describing urban resilience from a relational meaning perspective by starting with space syntax theory, which considers space as the dominant factor, and then superimposing other resilience mechanisms—social, economic, and environmental.

Changing pedestrian and vehicular evacuation behaviors along road networks can improve local disaster preparedness. Chang H.S. and Liao C.H. used a GIS and space syntax to develop an assessment model to determine where to locate safe and convenient emergency shelters based on evacuation behavior during mobility-based evacuation to improve the safety and mobility of the road network index. The best situation resulted in 35.3% improved efficiency [38]. Camilla Pezzica, Valerio Cutini, and Clarice Bleil De Souza found that space syntax could be used immediately and provide rapid information for urban social and economic problems to orient post-disaster decision-making [39].

### 2.4. The Introduction of Space Syntax and the Benefits of Evacuation Planning

Recently, evacuation maps have mostly been planned according to the physical environment or functional properties. The increasing use of applications of space syntax for evacuation plans for routes and shelters in an area saves considerable time and reduces the need for surveys, and the pros and cons of conditions can be determined by the planner [36]. Thus, space syntax, developed by Bill Hillier (ULC, The Bartlett), was used in this study. Many of these difficulties are directly addressed by the research tradition known as space syntax, which employs analytical methods to study the cognitive levels of urban space, particularly the knowledge that underpins architectural and urban design [2]. Bill Hillier, A Penn, J. Hanson T. Grajewski, and J. Xu correlated urban spatial configurations to movement patterns as a measure of the global properties of the grid using a space syntax measure of “integration ”called RN, and it was consistently found to be the most important indicator [2]. A key property of interest is how the various configurational variables are distributed in the urban grid. This can be shown graphically by drawing “core maps” of the 10% most integrated lines in a system. In most towns or urban areas, integration core maps pick out the main thoroughfares and high-density areas (integration cores) to offer a graphic realization of the morphological efficiency structure pattern of a town or urban area. Further, Kaili Dou and Qingming Zhan performed an accessibility analysis of urban emergency shelters by comparing the Gravity Model and space syntax [40]. The result of Dou and Zhan’s research pointed out that space syntax’s local integration is superior to global integration in measuring place accessibility. Marcelo Cando-Ja’come, Antonio Martinez-Grana, and Virginia Valdes followed the research frame to study what would happen if an earthquake greater than 6 or 7 moment magnitude (MW) occurred in Quito, the capital and largest city of Ecuador. They determined the movement patterns and traffic flows of the population using graphs of spaces interconnected by streets to predict the spatial behavior of humans and their concentration in the mentioned sites. The results were integrated, and the population in built areas was found to have high to very high displacement and an intense population concentration [41]. M. Cando- Ja’come et al. determined that efficient transport roads should be within 0.8 RN or more in their research completed using space syntax theory. Furthermore, X. Zhang, A. Ren, L. Chen, and X. Zheng measured the accessibility of road networks in 36 major cities in China [42]. Zhang’s huge survey showed that the high accessibility level was about 1.3972~1.7887 RN in cities such as Zhengzhou, Xi’an, Shijiazhuang, and Beijing. This study attempted to use accessibility to upgrade disaster prevention by using objective analysis methods and interpreting data from geometric mathematical analysis.

Further, the road in one area was regarded as the spatial unit based on quantity indicators, such as accessibility, choice, connection, control, and mean depth, and used to discuss the topology configuration and relationship between axial and visibility factors [39]. In addition, the importance of each unit was ordered by the topological view and compared with the actual location of the shelter to realize the difference in the results of digital tools and planners [38]. The aim was to improve disaster prevention in inefficient areas. The aims of our research on disaster prevention in this study were as follows:To reduce survey and labor work and establish a way to evaluate the efficiency of networks.To control and enforce important roads using an integration ranking to ensure the width of roads in the event of a disaster.To determine the best path for rescue, with the smallest distance and time to arrive at shelters.

### 2.5. The Influence of Axial and Accessibility Factors on Evacuation

To evacuate in a timely manner, prevention plans of each village are mainly based on a temporary shelter within 350 m for people to walk to and coordinated with factors such as junior high schools, elementary schools, residents, and the density of buildings in the area [7]. Therefore, the management of the evacuation area can be easily carried out.

Residents can go to the temporary refuge quickly in the evacuation area when disasters occur. After rescuers arrive or aftershocks occur, the residents are guided to the designated safe shelters. The temporary refuge at this level accepts people who cannot access safe shelters directly [38]. Stays are for the short term, and long-term refugees who are waiting for rescue in the temporary shelter are guided to the higher-level shelter. After aftershocks occur, people wait for the next step, depending on the situation. Temporary refuges are designated as neighborhood places such as parks, activity centers, junior high schools, and elementary schools.

Thus, besides the influence of the axial aspects of vehicles, the “visibility of pedestrians” is the most important factor for the evacuation routes of the village [36].

During the reaction to evacuation period, if the shelters are located at visible places along the city road networks, people can evacuate to the shelters and wait for rescue voluntarily. The time of evacuation is reduced due to the human labor and signs required. Additionally, during the rescue period, rescue workers can reach the shelter easily and help people reach long-term shelters or evacuate. Hence, highly visible and recognizable shelters and exits help people to evacuate when disasters occur.

### 2.6. The Evaluation Indicators of Space Syntax

Space syntax has generally been used in the research of urban morphology patterns [2,10,12]. The efficiency and availability of spatial systems are realized by analyzing the “integration” of roads [17]. In addition, the main route and connection of the spatial system are realized by analyzing the “choice” of routes [12]. Therefore, to create functional evacuation maps for Nei-Hu District, the availability of a spatial system of roads was evaluated by space syntax in this study.

#### 2.6.1. “Integration” Indicator of Axial Analysis

“Integration” (RN) is the accessibility to movement. The comparative value representing the accessibility of the location was obtained by comparing the mean values of the shortest route (i.e., relative depth) for one unit to the other in the road system. A higher value of RN for one unit means the location is more accessible, and the unit’s location is highly efficient and accessible in the road system. In short, a highly accessible route means the core section of the road in the system reaches the other road with the fewest corners. Higher and lower accessibility are represented in red and blue, respectively, in the software.
(1)$$\mathrm{Formula}:\;Rn=\frac{1}{RRA},RRA=\frac{RA}{DK}$$
where *Rn* is the relative accessibility of the whole district, *RRA* is the real comparative value of asymmetry, *RA* = 2(*d* − 1)/*k* − 2 is the value of asymmetry, *DK* is the assumption of the relative depth of “symmetry”, *d* is the mean relative depth of the location, and *k* is the number of units, *n* − 1.

#### 2.6.2. “Choice” Indicator of Axial Analysis

“Choice” is the degree of choice exercised through movement. Choice was quantitatively analyzed as the connection between one unit and a neighbor unit, dividing the relative depth in space equally.

A high number for a section of road passed through frequently represents that the unit is used repeatedly by choice compared to other routes in the system. In brief, a high choice value for a route means it is a necessary route of connection in the road network. If the route is broken, the other route cannot be passed through. Therefore, the system can determine the necessary sections of road. The section of road that is the most rapid and shortest can be determined. Additionally, a road that cannot be broken can also be determined to reduce the island effect when disasters occur. Higher and lower choice values are shown in red and blue, respectively, in the software.
(2)$$\mathrm{Formula}:\;Choice=\sum \;j\;\sum \;k\frac{d_{jk}\left(i\right)}{d_{jk}}$$
where *Choice* is the frequency of choice exercised through moment, *d_jk_*(*i*) is the shortest path from unit *j* to unit *k*, *d_jk_* is the shortest path from unit *j* to unit *k,* and the unit *i* (i.e., the unit itself) must be involved in the path for one section of the route.

#### 2.6.3. “Mean Depth” Indicator of Axial Analysis

“Mean Depth” (MD) is calculated by assigning a depth value to each space according to how many spaces it is away from the original space, summing these values, and dividing the result by the number of spaces in the system less one (the original space). The MD also refers to the mean depth of the route of each unit reaching all the other connecting routes. In the physical environment, bends, sections, and crossroads are regarded as independent units that result in one relationship of topological connection. The higher the MD value of the beginning point of the route and the relative route of the unit, the greater the number of bends and sections that are hard to reach or be used are passed through from the start of the route in the road network to any other relative location of the unit. Briefly, the relationship between construction and the road network is inaccessible. Oppositely, the lower the MD value, the more accessible the road network. The levels of mean depth are shown in the software.
(3)$$\mathrm{Formula}:\;MD\left(Mean\;Depth\right)=\frac{\sum_{j=1}^{n}\;d_{ij}}{n-1}$$
where Unit *i* is the beginning point of the root, *d_ij_* is the shortest route from unit *i* to unit *j* calculated by BFS, and *n* is the number of all units in the road network.

#### 2.6.4. Software Operation Example for Space Syntax Analysis

Higher and lower accessibility are shown in red and blue, respectively, in the software, as demonstrated in the following steps (Figure 3).

### 2.7. The Comparison and Evaluation of the Physical Environment—Hazard Risk Related to Building Age and Population Zoning

Finally, this study used a quadrant diagram, a scatter plot that divides the background into four parts (Figure 4). The *X*-axis (population) and *Y*-axis (buildings’ age) were used to draw the data containing two measurements as follows: the measurement target was used for environmental investigation in the quadrant diagram as a hazard measure tool [36]. In disaster management, the quadrant diagram is commonly used as a statistical method since the quadrant can represent the integration of two measurement definitions. Both the measured indicators were higher than average and can be placed in the first quadrant, which is the region with the greatest population cluster and oldest building age; broadly, we can define it as a relatively high-risk region. One of the X- or *Y*-axes was above average and could be placed in the second quadrant or the fourth quadrant, which was a loose zone. However, both of the measured indicators were below average and could be placed in the third quadrant, which was the lowest-risk area. Of course, disasters happen unpredictably, but in this study the issue of resource allocation and safety management in the area of the third quadrant was rethought due to the high risk of disaster management relative to the overall urban system [5,6,8].

## 3. Results

The spatial efficiency of the road network in Nei-Hu District analyzed by space syntax had obvious topology features in this study. Based on the ranking of accessibility and efficiency for route units, the units with higher integration in the new region (the feature is geometric topology) and with lower integration in the old region (the feature is organic topology) were concentrative layouts. The villages in the new and old regions were sampled randomly and separately for axial and visibility analysis, as outlined in the following sections. Whether the function of the road network matched the location of shelters was not evaluated. The cause of the difference in the function of the evacuation path in the new and old regions was evaluated and surveyed by space syntax.

### 3.1. Evaluation of the Prevention Map: A Case Study with Whole-District Axial Analysis

The roads in Nei-Hu District were evaluated according to the RN indicator. The new regions were developed recently, and the topology shape was geometric. The distribution of the routes with high integration analyzed by space syntax was concentrative. However, the old regions were developed early, and the topology shape was organic. The road system with low integration should be allocated and supplied efficiently with various policies and practices for disaster prevention plans [43].

Figure 5 features RN value analysis mapping and choice value analysis of the road network, which are useful for understanding the operation mode of evacuation paths. For example, the high-integration paths created a “crossroad” topology shape in the high-accessibility area and a “horizontal and vertical” topology shape in the areas with a clustered distribution. However, while these two pathways had high accessibility characteristics, they were not classified as emergency evacuation routes. Using space syntax analysis, the RN Values of path-1 were determined to be 1.730, 1.649, and 1.580 (choice indicator was 24004), and the path-2 values were 1.267 and 1.755 (choice indicator was 14906), as shown in Figure 6 below. They had the same efficiency characteristics in urban evacuation plans [36,40,41,42]. In terms of prevention plans and risk management, these paths are the main priority in terms of spatial structure when people try to escape or move. This study focuses on what causes these paths to be more efficient, as well as determining their associated urban patterns. By connecting and configuring, these paths converged and had a high degree of hierarchy in the physical environment. Therefore, keeping these paths operational, functional, and efficient means of guiding people to official shelters is the priority for disaster management in the reduction, preparation, reaction, and rebuilding stages. Further, it is necessary to ensure and maintain the width and length of integrated paths to establish a carrying capacity for disasters.

In contrast, there are some routes that are not classified as evacuation routes but are highly efficient, and whether these routes can contribute to the distance or time of disaster prevention and evacuation is the focus of this study. Additionally, we aim to prove that the accessible road can effectively shorten the distance of the escape and evacuation path and serve a wider range of places; follow-up must analyze the path and distance for each neighborhood.

Nevertheless, the two roads that were not 20 m wide were not regarded as emergency roads in the original prevention plan, as shown in Figure 6. Based on the analysis by space syntax, if two roads could be added to evacuate, the connecting function of paths could be increased and upgraded for remote areas. Then, the low-efficiency areas that are not reached would be improved. Therefore, this paper suggests that the original whole-district evacuation map should be supplemented and modified to be more safe by space syntax, as shown in Figure 6.

As seen in the above table, the original and increased path distances of the 39 neighborhoods showed very different effects (Table 1). First, the average evacuation distance becomes shorter. Second, the shortened path length is 157.56 m, and the evacuation effect is increased by 16.82%. Third, the average time of two and a half minutes is shortened to three minutes for adults and four to five minutes for children. This is an empirical study of many effects. This study proved that the accessibility path analysis of space syntax is valuable, and it also provides a high degree of contribution to the field of geometric distance research.

In previous literature, Leu Chiang-Hui proposed a route selection model in an urban earthquake disaster relief study to improve transportation efficiency by 33% [32]. Hou Peng-Hsi established a survivable network design model for earthquake disasters to decrease the average travel cost by 33% [33]. Chang H.S. and Liao C.H. used a GIS and space syntax to develop an assessment model to determine where to locate safe and convenient emergency shelters based on evacuation behavior during mobility-based evacuation to produce a safety and mobility index of the road network. The best situation resulted in 35.3% improved efficiency [38]. The reason this study shows relatively low efficacy is related to the real environment or geographical location, which is analyzed in the next section.

### 3.2. Combination with Accessibility Mapping and Physical Environment

A quadrant diagram is a scatter plot that divides the background into four parts (Figure 7). The *X*-axis (population) and *Y*-axis (buildings’ age) were used to draw the data containing two measurements as follows: the quadrant plot presented a completely uniform distribution with 12 units in the first quadrant, 12 units in the second quadrant, 8 units in the third term, and 7 units in the fourth quadrant. The first quadrant is a relatively highly concentrated area with a large population, old houses, and high disaster risk. Of course, in a disaster, whether it is an earthquake or a natural disaster, there are many people who need to escape or take refuge, and this is a relatively dangerous place. Old buildings are also a high-risk problem, especially during earthquakes, which can not only damage buildings and cause them to collapse but also block road connections in the city, affecting the movement of firefighters or police traveling to evacuated areas. In this study, the average population per mile in Neihu District was 6961, and the average age of houses was 26.3 years.

After classifying the four quadrants, we found a high degree of characteristics. Additionally, it was highly similar to the previous research analysis; that is, in the first quadrant area, the effect of increasing the vacuum particle size distance was as high as 31.61% (Table 2). This is an amazing study result. This result not only shows similarity to previous studies but also directly proves that the high-accessibility path is helpful for high-hazard-risk neighborhood areas. This is important for developing an effective understanding and creating analyses in government departments or disaster management. In terms of urban governance, resources are limited. Rather than dedicating new resources, high-risk areas in the existing urban landscape should be adjusted or centrally managed. This is also the most important work in the disaster preparedness phase.

This study used quadrant maps in relation to accessible roads on a comprehensive map. We could clearly see that since the areas in the first quadrant were relatively close to accessible roads, and there was an area that was not connected to the existing evacuation channels, the increased channels became more accessible and extensive (Figure 8). Such suggestions help government departments prepare for disaster management.

## 4. Discussion

### 4.1. Spatial Features in Accessibility Mapping

The road network of accessibility, in the case of this study, is highly characterized by clustered areas. Such an analysis model is helpful for the management of urban disaster prevention. In addition, for managing urban growth, it can also help to understand the pattern from the perspective of route evacuation. The four items that influenced disaster prevention were whether the route of the shelter was accessible, whether the shelter was visible, whether the route and visibility were matched and recognized, and the mean depth of the road system. These factors will be compared to determine the most helpful spatial model for disaster prevention planning. In addition, unhelpful factors of disaster prevention will be discussed [44].

Based on a report of massive earthquakes in Taiwan, such as the 921 earthquake, 91% of people escape from the building to the emergency road in front them and find a place near their house for evacuation within half a day of the earthquake’s occurrence. They are settled temporarily within half a day to two days. Residents who cannot go back to their house for more than two weeks are accepted in the shelter for a long period of time. The number of evacuated people is greatest within half a day to three days. The characteristics of popular shelters were generalized as follows: they were places where people could wait for rescue near their houses, they were open areas that made people safe, and they were familiar environments [8]. In brief, people generally prefer a place for sheltering that can be recognized well, which falls under the behavior model of temporary evacuation.

The spatial characteristics of the physical environment were explained by the accessibility and efficiency of geometric analysis by space syntax. This paper presents a high-efficiency accessibility value, the same result as M. Cando- Ja’come’s [41] and X. Zhang’s [42] research. It can be concluded that space syntax is suitable for application as a spatial accessibility analysis tool in urban research topics. Moreover, the mean depth of the road network was determined by space syntax. Roads with many turning points resulted in the complication of axial and visibility factors. Axial and visibility characteristics with road turns are hard to find and pass through. Spatial recognition is also made difficult. It takes people more time to use roads without guidance by signs or directions by guides, which influences the key times for evacuation. The model makes the distance and time for evacuation more dangerous in the physical environment.

### 4.2. Surveying Evacuation Map’s Efficiency Patterns, Configured by Axial and Visibility Factors

Using space syntax, the overarching conclusion of this paper proved some discrepancies between biological planners and digital computers [40]. It can be concluded that a hidden gap exists between urban planners and artificial intelligence, and space syntax can be used to plan more safe and efficient prevention plans. In the case of unpredictable hazards, this paper contributes to ongoing urban prevention and establishes some path patterns and quantitative indicators to present to integrated or disintegrated communities. These indicators affect the time to escape and can be used by government administration. The main findings reveal that using space syntax to examine the prevention map has important implications for urban safety.

Based on the analysis of the cases mentioned above, urban patterns summarized in this study helpful for disaster prevention planning of efficiency networks were as follows, and plans should reflect these strategies:

Straighten the shape of routes

According to the shape of routes, straight roads were more helpful for evacuation than curved roads. The same results were obtained in the whole district and local areas based on the accessibility of routes analyzed by space syntax. The connection between axial and visibility factors is helpful for evacuation planning. Moreover, the important route is formed by the influence of connection and convergence simultaneously. Whether the road is wide enough to evacuate the required number of people is the next issue that should be faced and ensured. Nevertheless, based on the results obtained in this study, a road of straight travel has the advantage of good evacuation. The problems of management for evacuation should be managed later on [42].

2.Decrease the number of turns

Decreasing distance and saving time increased spatial recognition and were both positive indicators influencing evacuation. In the cases analyzed in this study, too many turns influenced accessibility and efficiency and decreased the function of the road system. The road configuration for disaster prevention planning was determined in this study. Roads with too many turns were not a good urban model. Based on the cases of new areas in this study, we found that the geometric system of urban planning was an efficient configuration with apparent proof. Moreover, according to the route analysis of the whole district, some roads not wider than the outside ring roads were more efficient for passage, as analyzed by space syntax. The outside ring roads that were sectioned or curved due to the landform were less important in the analysis of the whole district [45].

3.Zone by communities, not administrative villages

By studying the correlation between the physical environment and space syntax, this study discovered that a village assigned one shelter is a good policy and also found a weakness in that the shelter has to be arranged in the middle location or the best site in the physical environment. For example, the situation in Nei-Hu Dist. in Taipei City is that some neighborhoods are close to the next village’s shelter in the physical environment but far away from the official shelter on the evacuation maps. This study addressed cases where people from neighborhoods located at the village boundary had to go through the front road, not the main road or the most integrated one, which leads to a long distance to shelters. However, this study offered theoretical and empirical evidence for the need to reform the zoning instead of “emergent living fields” by communities or neighborhoods, not the administrative village. In terms of urban prevention plans, this article concludes by discussing the implications of this study for assessment based on the physical environment to ensure effective function during disasters [41].

## 5. Conclusions

In Taiwan, the “Urban Disaster Prevention Planning Manual” published by the government indicates that the evaluation of the prevention function of evacuation shelters is a priority for further research [7]. Another thesis also showed these issues by studying correlations between the location of shelters or aid stations and the service area of the road network, as well as post-occupancy evaluations of “emergent living fields” using situation, service capacity, supply area, and complaint investigation. To optimize evacuation plans for urban planning [8], this study developed and explored these issues in terms of topology configuration topics, focusing on accessibility and visibility of urban streets and road networks with the quantitative analysis of axial and visual factors. “Space syntax” is a valued analysis tool that can be used to rank streets to determine the most efficient paths for people’s evacuation. The aim of this study was to make paths more convenient and identify them in urban space, as well as create some suggestions for evacuation maps as follows:

### 5.1. Reforming Evacuation Plans with the Values of Axial Efficiency and Accessibility

Based on the analysis of the cases mentioned above, the configuration of urban patterns summarized in this study helpful for disaster prevention planning with an efficient network was as follows. Shortening distance and saving time increased spatial recognition and were both positive indicators of evacuation. In the cases analyzed in this study, too many turns influenced the accessibility and efficiency and decreased the function of the road system.

The road configuration for disaster prevention planning was determined in this study. A road with too many turns was not a good urban model. Based on the cases of new areas in this study, the geometric system of urban planning was found to be an efficient configuration with apparent proof. Moreover, according to the route analysis of the whole district, some roads that were not wider than the outside ring roads were more efficient for passage, as analyzed by space syntax. The outside ring roads that were sectioned or curved due to the landform were less important in the analysis of the whole district.

By studying the correlation of physical environment and space syntax, this work discovered that a village assigned one shelter is a good policy and also found a weakness in that the shelter has to be arranged in the middle location or best site in the physical environment. For example, the situation in Nei-Hu Dist. of Taipei City is that some neighborhoods are close to the next village’s shelter in the physical environment but far away from the official shelter on the evacuation maps. This study addressed cases of people in neighborhoods located at the village boundary who have to go through the front road, not the main road or the most integrated, which leads to a long distance to shelters. However, this study offered theoretical and empirical evidence for the need to reform the zoning instead of “emergent living fields” by communities or neighborhoods, not the administrative village. In terms of urban prevention plans, this article concludes by discussing the implications of this study for assessment based on the physical environment to ensure effective function when disaster strikes.

### 5.2. Further Research

Disaster prevention planning was considered from the planner’s point of view and focused on voluntary evacuation during periods of evacuation. The model of the road influences the values of axial efficiency and visibility, as proved by this study. Evacuation behavior with better configuration would be beneficial to usage and management late on and increase the time allowed for people to react to disaster. However, which behavior was important was not the issue in this study. According to urban planning, the axial accessibility and visibility would be more efficient with a geometric space distribution. Consequently, disaster prevention should be evaluated as follows:Conduct a social or economic materiality query and survey.For buildings on the evacuation route, conduct on-the-spot interviews or segmental records; after all, different structural forms or heights have different degrees of danger.Distribute disaster rescue support. After all, after a disaster occurs, the start of rescue is a very important phase. It is also necessary to add scenario simulations to analyze the differences in rescue or evacuation modes in response to different types of disasters.

## Figures and Tables

**Figure 1 ijerph-20-02913-f001:**
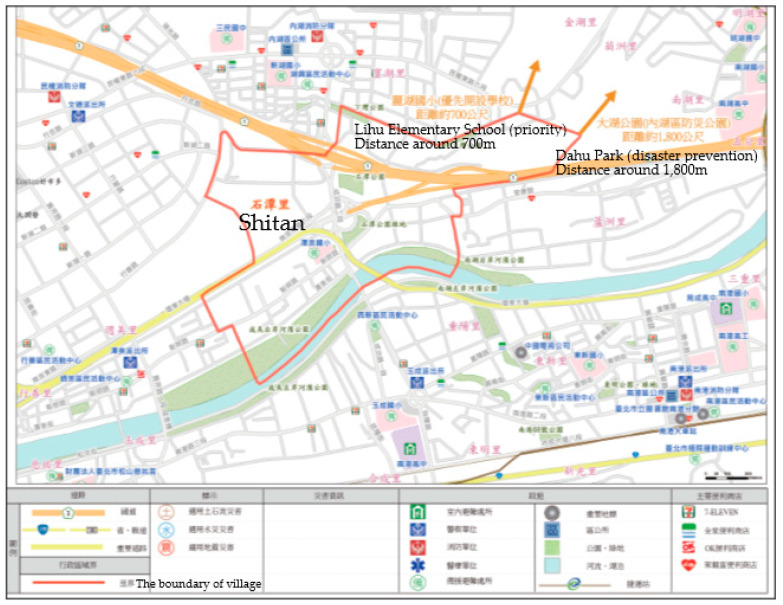
A village’s evacuation map in Nei-Hu District. Source: village evacuation map by the Taipei City Government.

**Figure 2 ijerph-20-02913-f002:**
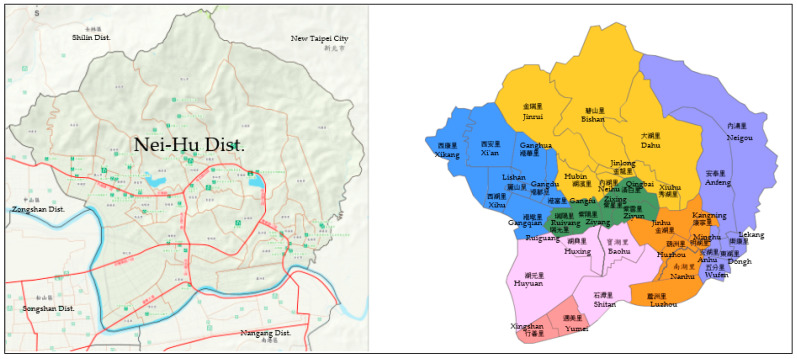
Nei-Hu District disaster prevention map for 39 neighborhoods’ evacuation routes. Source: prevention map by Taipei City Government.

**Figure 3 ijerph-20-02913-f003:**
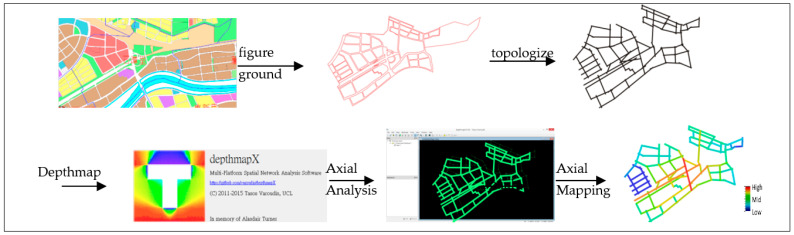
Software operation example.

**Figure 4 ijerph-20-02913-f004:**
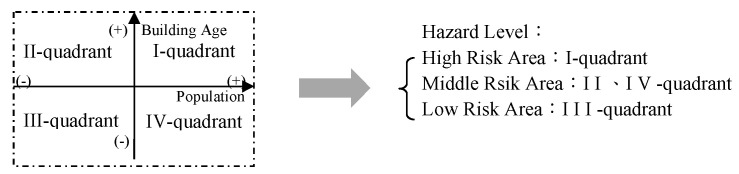
Quadrant diagram of hazard risk level.

**Figure 5 ijerph-20-02913-f005:**
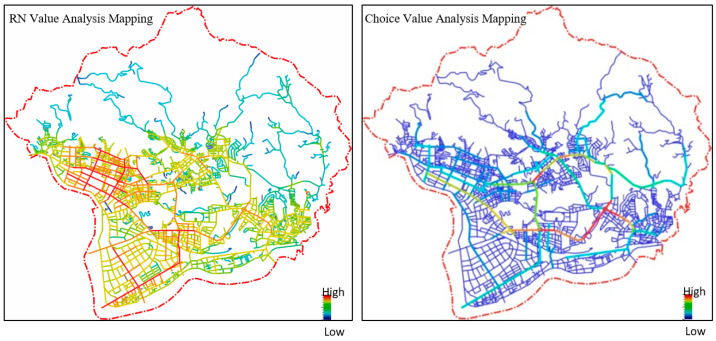
The whole-district axial analysis map of Nei-Hu.

**Figure 6 ijerph-20-02913-f006:**
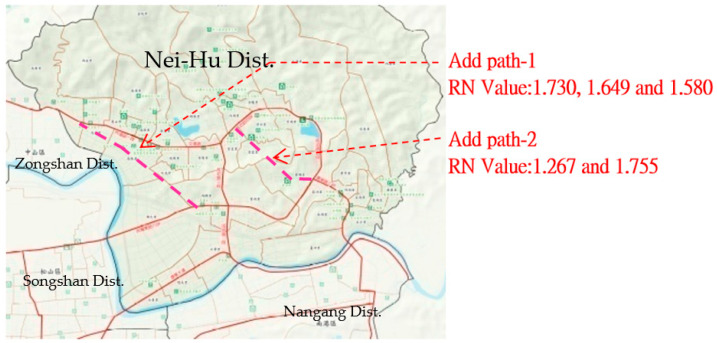
Added accessibility path.

**Figure 7 ijerph-20-02913-f007:**
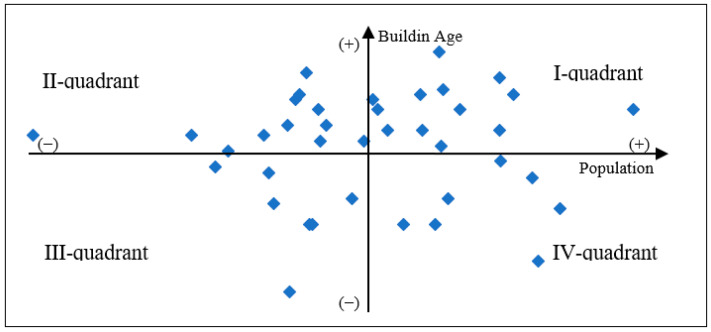
The 39 neighborhoods of Nei-Hu District: quadrant diagram.

**Figure 8 ijerph-20-02913-f008:**
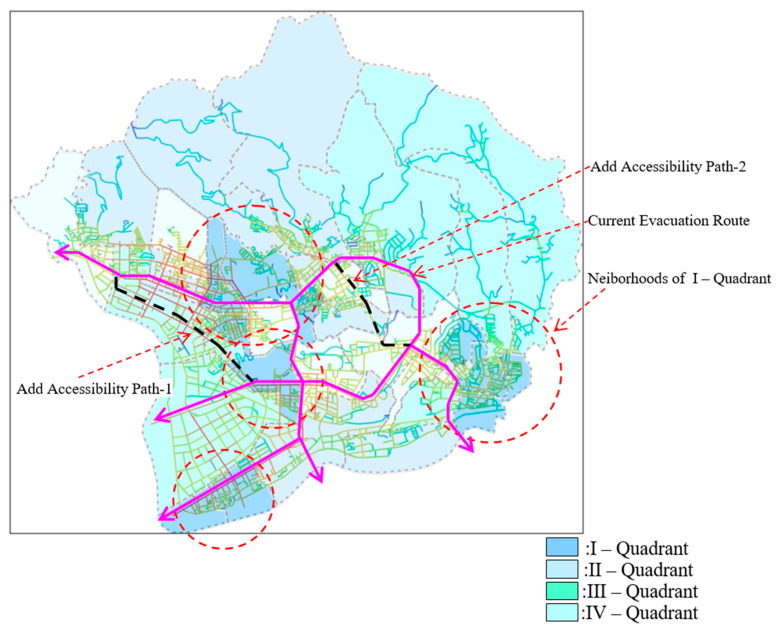
Comparison mapping of added accessibility path.

**Table 1 ijerph-20-02913-t001:** Comparison chart of distance to evacuation and accessible road.

No.	Index-1	Index-2	Index-3	Index-4	No.	Index-1	Index-2	Index-3	Index-4	No.	Index-1	Index-2	Index-3	Index-4
1	725	570	6	5	14	390	335	4	3	27	1750	1720	10	9
2	500	375	7	6	15	430	335	3	3	28	390	300	3	3
3	590	390	4	4	16	750	355	5	4	29	2650	2500	21	20
4	735	510	4	3	17	2620	2310	47	44	30	1150	825	4	4
5	530	425	7	5	18	2750	2620	22	22	31	1950	1900	4	4
6	700	370	8	7	19	1070	1020	5	5	32	790	570	5	5
7	590	395	4	4	20	1030	920	6	6	33	675	550	6	5
8	720	590	5	4	21	775	720	8	7	34	750	575	6	6
9	990	790	4	4	22	395	290	5	5	35	825	700	5	4
10	730	495	6	5	23	1200	1120	6	6	36	910	795	7	6
11	550	290	5	4	24	530	350	4	3	37	595	500	5	5
12	970	500	11	9	25	500	350	5	5	38	1100	1050	6	5
13	1120	950	14	14	26	525	520	4	4	39	570	535	13	12

Average (index-1 − index-2) = 157.56 m. Percentage (index-1 − index-2)/(index1) = 16.82%. Average decrease in the number of turns: −0.67 times. Note: Index-1: Distance to evacuation road (meters). Index-2: Distance to accessibility road (meters). Index-3: Turns to evacuation road (no. of times). Index-4: Turns to accessibility road (no. of times).

**Table 2 ijerph-20-02913-t002:** Comparison chart of four-quadrants diagram.

Quadrant	Numbers	Ave. of Index-1	Ave. of Index-2	Subtract	Percentage
I	12	695.58 m	475 m	219.58 m	31.61%
II	12	1088.33 m	940.83 m	147.5 m	13.55%
III	8	1213.8 m	1985.6 m	127.5 m	10.51%
IV	7	775 m	672.14 m	102.86 m	13.27%

## Data Availability

Not applicable.

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
