# Peer review of "Contribution of Accessibility to Urban Resilience and Evacuation Planning Using Spatial Analysis"

_ijerph, 2023, doi:10.3390/ijerph20042913_

Round 1
Reviewer 1 Report
Dear Authors
The manuscript entitled "Accessibility to Contribute to Urban Resilient and Evacuation Planning by Spatial Analysis" presents a relevant theme in socio-environmental terms, incorporating elements of human life, the conditions of environmental disasters, and human work for the maintenance and protection of life from the perspective "Space Syntax Topology indicators." The authors present the study's contribution and the complementation that it produces concerning the approaches most used and the original contribution of the present study. They use complex theories of the humanities and simultaneously with the tools of the theory of Spatial Synthesis.
I made small suggestions in the method, in the presentation of results and discussion, to maintain consistency in the thematic language and structure of the text.
Introduction:
1- Briefly present the general socio-demographic characteristics of the two regions categorized as old and new. Alternatively, these characteristics could be included in the discussion. They can intensify behavioral elements of contexts presented in the discussion.
Results:
2- I suggest improving Figure 4 (Area C and Area D);
3- I suggest presenting a computational figure on the global Syntax of both urban spaces (new and old) in an explanatory way about the differences and similarities described (slightly different from figure 6).
Discussion:
Suggestion: 4 - The authors could expand the comments on "the road with good shape had the advantage of good evacuations naturally" (p.13), given the characteristics of the regions studied.
Thank you for reading in manuscript review. The manuscript's content expresses the authorship of scientific expertise to influence socio-political and environmental decision-making for preserving human life in natural disasters.
Author Response
Response to Reviewer's Comment:
1.I have modified the result and added the population and buildings age in 3.2 chapter.
2.figure. 4 has been replaced.
3.The features were discussed in 3.2 and 4.1 chapter.
4.The extend literature is written in 2.2, 2.3 and 2.4 chapter.
Your acknowledgement will be highly appreciated. Thank you
Sincerely your,
Author
Reviewer 2 Report
This study used “Space Syntax” to carry out urban evacuation planning, assessed the evacuation efficiency of the road system during an evacuation, and improved evacuation safety, providing new insights for issues related to urban evacuation. However, the following related issues still need to be clarified:
1. Related studies on the use of Space Syntax and how they differ from this study should be included.
2. There are numerous studies on urban evacuation. A comparison of the results and the different tools used should be briefly explained.
3. The Chinese explanations in Figure 1 should be changed to English.
4. The reason for setting the location should be included and explained in the form of a prerequisite.
5. An “evacuation site” is also an extremely important topic in urban evacuation planning. Explanations on the setting as well as the simulation results of this study should be added.
6. Section 2.2 mentioned that the distance and time of an evacuation will be investigated in this study; however, it seems that there is no result for such an investigation.
7. The result for “Area-A and B” is completely different from that of “Area-C and D,” which is a well expected result. Does the result have a close connection with the topography? Please add your explanation.
8. The key to evacuation time in a building evacuation is “the time required to pass through an exit.” However, there is no such problem in urban evacuation. What are the key factors affecting urban evacuation? Does this study touch upon these issues?
9. The conclusion explained the impact of road systems on evacuation, which is consistent with concepts that are commonly known. Are there any other new insights?
10. Different methods can always bring about different perspectives or academic advancement; how to effectively apply those methods is another topic for discussion. For the conclusion, it is recommended that more suggestions be provided for the government sector.
The Space Syntax theory has been widely used in architecture and urban related issues. However, its promotion is not easy due to being limited by the high entry barrier in software use. We look forward to the follow-up work of this study.
Author Response
Response to Reviewer's Comment:
1.The chapter is written in 2.4.
2.The literature has been summary in 2.3 chapter.
3.All words are confirmed in English.
4.The reason has been discussed in 3.2 chapter.
5.This part is summed up to further research in 5.2 chapter.
6.I have added this analysis in 3.1 chapter ( lines 458 ).
7.The result is re-written and added new table to discuss the finding.
8.The result is re-written and added new table to discuss the finding.
9.This paper response to other literature and presented the similar results.
10.This part is summed up to further research in 5.1 chapter.
Your acknowledgement will be highly appreciated. Thank you
Sincerely your,
Author
Reviewer 3 Report
Although the research motivation of this article is correct and good, but in terms of research rigor, data reliability and graphic quality, it still does not fully meet the basic requirements for journal publication, and it can still be used as a general assignment report or conference article. However, this article is still unable to obtain review permission for publication as a journal article. The review comments are as follows:
1. The author uses the analysis method of space syntax theory as a reference for urban disaster prevention routes, but there are too few references to the research on space syntax applied to urban design and evacuation, and the overall research is only for reference, not based on field observations with validation.
2. The quality of the overall figures and tables in the article is too poor and the meaning is unclear, such as Figure 1. in line 74, Figure 2. in line 77, Figure 4. in line 309, Figure 6 in line 319, etc. Image quality and explanatory None of them are sufficient to meet the basic publishing standards of international journals.
3. Table 1. A chart of local village Axial and Visibility analysis maps in high efficiency areas (new 344 region) in lines 343~345. 386-388 Table 2. A chart of local village Axial and Visibility analysis maps in 386 low efficiency areas (old 387 region). The visual visibility is described in the table. The street profile is used as the base map, which does not conform to the actual visual visibility range of the user. The building configuration should be used as the basic base map to meet the actual visual obstruction state. In the part of the axial line diagram, the author did not explain the analysis difference between the evacuation line and the urban vehicle road network, the principle of grading, and even the analysis principles of the road infrastructure, such as the facilities between pedestrian passages and vehicles. None are defined.
4. Line 400, Table 3. A table of spatial features in village evacuation maps. The evaluation mode of right and wrong is too simplified, and the specific meaning of quantitative values is not explained. Such as (415-419)"; (1) the distance could be shorten with accessibility of routes. (2) the areas with wide visibility should be open 416 enough. (3) it would be good for spatial recognition and helpful for people to feel417 belonging to the environment if the axial and visibility matched together. The spatial characteristics of physical environment were explained by the accessibility and efficiency of axial and visibility analyzed by Space Syntax.". Generally speaking, it is well known that having good visual and mobile accessibility can help evacuation behavior, but it does not explain how much the axial line accessibility and visibility must be to explain the method of article is feasible.
Author Response
Response to Reviewer's Comment:
1.The extend literature is written in 2.2~2.5 chapter.
2.All the figures are modified for better quality.
3.The result is re-written and added new table to discuss the finding, like 3.1 chapter.
4.The reason has summed up in 3.2 chapter.
Your acknowledgement will be highly appreciated. Thank you
Sincerely your,
Author
Round 2
Reviewer 3 Report
After adjusting the figure quality of this paper, it can be reviewed as an accepted journal article. It is suggested that the analytical value of space syntax must be further explained. For example, how much visual visibility must be achieved as a shelter is suitable. In addition, the relevant literature on the application of space syntax to evacuation research still needs to be supplemented and cited. After completing the above requirements, it can be reviewed as an accepted article.
Author Response
Dear Reviewer
Thanks for your wisely comments for this research article. After adjusting the methodology citations and explanation of accessibility value, the clarity of the main issue in this paper became a high quality level by your suggestions, and revision is modified as following:
1.The application of Space Syntax literature is cited as no.33~43 references, and their research summary is discussed in 2.3 and 2.4 chapters. These literatures issued the spatial configuration influence the accessibility in urban system in geometric analysis argumentation. Particularly, the accessibility level distributed more than 0.8 value to efficiency roads in no.41 citation, and the no.42 addressed the high accessibility level is about 1.3972~1.7887 value of 36 major cities in China.
2.Excitingly, we also found the same result is 1.58~1.73 and 1.267~1.755 value in 3.1 chapter of this paper. And the correlation in 1rt quadrant, that is a high risk area, extends to 31.61% accuracy in 3.2 chapter. This is delighted to contribute in spatial research analysis field.
Sincerely,
Authors